# Epidemiology of Antibiotic Resistant Pathogens in Pediatric Urinary Tract Infections as a Tool to Develop a Prediction Model for Early Detection of Drug-Specific Resistance

**DOI:** 10.3390/antibiotics11060720

**Published:** 2022-05-26

**Authors:** Francesca Bagnasco, Giorgio Piaggio, Alessio Mesini, Marcello Mariani, Chiara Russo, Carolina Saffioti, Giuseppe Losurdo, Candida Palmero, Elio Castagnola

**Affiliations:** 1Scientific Directorate, IRCCS Istituto Giannina Gaslini, 16147 Genova, Italy; 2Division of Nephrology, Dialysis, and Transplantation, IRCCS Istituto Giannina Gaslini, 16147 Genova, Italy; giorgiopiaggio@gaslini.org; 3Infectious Disease Unit, Department of Pediatrics, IRCCS Istituto Giannina Gaslini, 16147 Genova, Italy; alessiomesini@gaslini.org (A.M.); marcellomariani@gaslini.org (M.M.); carolinasaffioti@gaslini.org (C.S.); giuseppelosurdo@gaslini.org (G.L.); eliocastagnola@gaslini.org (E.C.); 4Department of Neuroscience, Rehabilitation, Ophthalmology, Genetics, Maternal and Child Health (DINOGMI), University of Genoa, 16132 Genova, Italy; 5Infectious Diseases Unit, Ospedale Policlinico San Martino IRCCS, 16132 Genova, Italy; chiara.russo16@icloud.com; 6Clinical Pathology Laboratory and Microbiology, IRCCS Istituto Giannina Gaslini, 16147 Genova, Italy; candidapalmero@gaslini.org

**Keywords:** urinary tract infections, pediatrics, antibiotic resistance, risk factors, prediction model

## Abstract

Antibiotic resistance is an increasing problem, especially in children with urinary tract infections. Rates of drug-specific resistant pathogens were reported, and an easy prediction model to guide the clinical decision-making process for antibiotic treatment was proposed. Data on microbiological isolation from urinoculture, between January 2007–December 2018 at Istituto Gaslini, Italy, in patients aged <19 years were extracted. Logistic regression-based prediction scores were calculated. Discrimination was determined by the area under the receiver operating characteristic curve; calibration was assessed by the Hosmer and Lemeshow test and the Spiegelhalterz test. A total of 9449 bacterial strains were isolated in 6207 patients; 27.2% were <6 months old at the first episode. *Enterobacteriales* (*Escherichia coli* and other *Enterobacteriales*) accounted for 80.4% of all isolates. Amoxicillin-clavulanate (AMC) and cefixime (CFI) *Enterobacteriales* resistance was 32.8% and 13.7%, respectively, and remained quite stable among the different age groups. On the contrary, resistance to ciprofloxacin (CIP) (overall 9.6%) and cotrimoxazole (SXT) (overall 28%) increased with age. After multivariable analysis, resistance to AMC/CFI could be predicted by the following: sex; age at sampling; department of admission; previous number of bacterial pathogens isolated. Resistance to CIP/SXT could be predicted by the same factors, excluding sex. The models achieved very good calibration but moderate discrimination performance. Specific antibiotic resistance among *Enterobacteriales* could be predicted using the proposed scoring system to guide empirical antibiotic choice. Further studies are needed to validate this tool.

## 1. Introduction

Urinary tract infection (UTI) is a common health problem in pediatrics [1]. The main targets in childhood urinary tract infection treatment are rapid recovery and prevention of related complications, such as urosepsis and renal abscess, and permanent renal parenchymal damage that can cause chronic renal failure and hypertension [2,3,4,5]. To achieve these aims, empirical antibiotic prescription is often endorsed already before the culture results are available, since a delay in treatment initiation could be associated with permanent renal scarring [6]. Administration of low-dose antibiotic prophylaxis is a common practice in the presence of recurrent urinary tract infections due to 3rd–5th degree vesico-ureteral reflux [7,8,9]. At the single-patient level, the efficacy of antimicrobial treatment is critically dependent on correctly matching the antibiotic(s) to the specific susceptibilities of pathogens [10,11,12]. However, the antibiotic resistance of urinary tract pathogens isolated in children is increasing, especially for commonly used antimicrobials. This has important implications for treatments [13,14]. For this reason, knowledge of the etiology of UTIs and their resistance patterns in general and in specific geographical locations may aid clinicians in choosing the appropriate antimicrobial treatments, reducing the risk of incorrect initial treatment, especially in cases of recurrent infections [10,14,15,16]. The common etiological agents of pediatric UTIs are Gram-negatives, mainly *Escherichia coli*, followed by *Klebsiella pneumoniae* and *Proteus mirabilis*; and *Enterococcus faecalis* among Gram-positives [1,17]. Resistance rates exceeding 20% for commonly used drugs have been described for these pathogens [17,18,19]. The risk of resistance to different antibiotics is associated with patient demographics; comorbidities; and patient’s clinical history, including recurrent UTIs, especially in the presence of malformations and prophylaxis administration, hospitalizations and antibiotic resistance in previous infections [11,16,19,20,21]. 

The aims of the present study were to describe the proportions of drug-specific resistant pathogens and to develop a simple, initial drug-specific risk prediction model of antibiotic resistant UTIs.

## 2. Materials and Methods

The Istituto Giannina Gaslini (IGG), Genoa-Italy, is a tertiary care pediatric hospital in Northern Italy serving as a local pediatric hospital for the Genoa area but also representing a referral hospital nationwide and for many foreign countries. 

Data on microbiological isolation from urinoculture, in the period January 2007–December 2018, in patients <19 years old, were extracted from the IGG Laboratory of Microbiology database and anonymously analyzed. For each episode, the following data were available: age at time of positive culture (considering median and interquartile range, it was categorized as: ≤6 months, 7 months–2 years, 3–7 years and >7 years), sex, number of previous episodes (0, 1 or >1) and department of admission where the UTIs were diagnosed.

All positive urinary cultures were considered, independently from a diagnosis of upper (pyelonephritis) or lower (cystitis) UTI or asymptomatic bacteriuria [1,22]. All urine samples were obtained by midstream clean-catch, catheterization or urine bags, according to patient’s age (continence or not) and clinical condition and international recommendations [23,24,25]. Samples were processed on Columbia agar +5% sheep blood (bioMérieux SA, Marcy-l’Etoile, France) and MacConkey agar (bioMérieux SA, Marcy-l’Etoile, France) and were incubated at 37 °C overnight. Significant growth was considered to be ≥105 colony forming units (CFU)/mL of urine. Urinary culture, independently from gender and isolated pathogen, was considered positive with ≥10^5^ CFU in all samples [22,26], since the method of urine collection was not available in our data [27,28]. The cultures with growth of more than two pathogens were excluded from the analysis because they considered as contaminated.

All antibiotic susceptibility tests were performed with automated systems (BD Phoenix, Becton, Dickinson and Company, Sparks, MD, USA) that were based on Clinical and Laboratory Standard Institute (CLSI) criteria from 2007 to 2010; but from 2011, European Committee on Antimicrobial Susceptibility Testing (EUCAST) criteria for definition of antibiotic susceptibility were adopted. Due to this major change and considering that EUCAST breakpoints could have been changed every year, we decided to report the results in terms of susceptibility or resistance according to the definitions provided by the system used each year, which corresponded to what was performed in everyday practice [19]. 

Isolated pathogens were grouped as: *Escherichia coli*, *Pseudomonas aeruginosa*, other *Enterobacteriales*, other non-fermenting Gram-negatives, *Enterococcus* spp. and other Gram-positives (details in Appendix A). For each isolated strain, antibiotic susceptibility to the following drugs was recorded: amikacin, amoxicillin-clavulanate (AMC), ampicillin, cefixime (CFI), cefuroxime, cefotaxime, ceftazidime, ciprofloxacin (CIP), fosfomycin tromethamine, gentamicin, meropenem, nitrofutantoin, cotrimoxazole (SXT) and piperacillin-tazobactam for *Escherichia coli* and other *Enterobacteriales*; amikacin, ceftazidime, ciprofloxacin, fosfomycin tromethamine, meropenem and piperacillin-tazobactam for *Pseudomonas aeruginosa*; amikacin, ceftazidime, ciprofloxacin, gentamicin, meropenem, and piperacillin-tazobactam for other non-fermenting Gram-negatives; and ampicillin, teicoplanin and vancomycin for *Enterococcus* spp.

### Statistical Analysis

Categorical data are reported in terms of absolute frequencies and percentages. Continuous data are described in terms of median values and interquartile ranges (IQR), due to their non-normal (Gaussian) distribution. Percentages of antibiotic resistant infections by pathogen were calculated, and the 95% confidence interval (CI) is reported with the cluster-robust estimator of variance allowing for intra-group correlation due to possible repeated UTI in the same patient.

The associations between the binary outcome variable (antibiotic resistance) and independent variables were assessed by logistic regression and are reported in terms of odds ratio (OR) and 95% CI. The cluster-robust estimator of variance was used, calculating CI to control for intra-group correlation. Multivariable regressions were focused on the group of Gram-negative (*Escherichia coli* and other *Enterobacteriales*) pathogens that are more homogeneous and representative in terms of tested susceptibility and frequency of antibiotic resistance. Available variables as independent factors were entered into the multivariable models, and the possible interactions between factors were also examined. The likelihood-ratio test (LR) was calculated to measure the effect of each predictor.

Prediction models were developed from the final logistic models to predict the risks of resistance to specific antibiotics (AMC, CFI, CIP and SXT) with oral formulations commonly used to treat Gram-negative UTIs in pediatrics. As multivariable analysis showed similar results of risk factors for AMC and CFI and for CIP and SXT, two variables were generated to consider AMC and CFI and CIP and SXT together. To correct for the optimism bias, data were randomized, using a random number generator, into either the training data set (70% of the total) or the validation data set. The models were fitted using the training data sets and then assessed by using the models to score the validation data sets. In addition, internal validation, k-fold cross-validation(k = 10), was performed. 

The overall predictive model performance was assessed using the Brier score (that includes component of discrimination and calibration); the predictive accuracy for discrimination was determined by calculating the area under the receiver operating characteristic (ROC) curve (AUC) and the discrimination slope; calibration was assessed by comparing predicted and observed probabilities using the Hosmer and Lemeshow test, the Spiegelhalterz test and the calibration plot [29]. To compare the areas under two independent ROC curves (training and validation) the Chi-square test was applied [30]. To develop the predictive risk score function, a weight was assigned to each risk factor in relation to each β parameter based on the multivariable logistic regression model [31]. Analysis and presentation were based on available data, i.e., no imputation of missing data was performed. All tests were 2-tailed, and a *p* value < 0.05 was considered statistically significant. All analyses were performed using Stata (Stata Corp. Stata Statistical Software, Release 13.1 College Station, TX, USA, Stata Corporation, 2013) and SAS software Version 9.4 (SAS Institute Inc, Cary, NC, USA). 

The TRIPOD statement recommendations for develop prediction models were followed [32]. 

This study was conducted in accordance with the Helsinki Declaration. According to Italian legislation, the study did not need ethical approval, as it was a purely observational retrospective study on routine collected anonymous data. Moreover, informed consent for participate in the study was not required since retrospective data were obtained by an anonymous microbiology database. In any case, consent to completely anonymous use of clinical data for research/epidemiological purposes is requested by clinical routine at the time of admission/diagnostic procedure.

## 3. Results

A total of 9449 bacterial strains were isolated in 9356 urine samples, Figure 1, from 6207 patients, and 1432 (23.1%) patients presented having had more than one episode (812 patients with 2 and 620 with >2). Half of patients were females, and the median age at the first positive sample was 1.8 years (IQR, 0.4–6.8); 1685 (27.2%) children were aged <6 months at the first episode. Characteristics of the study population are reported in Appendix A.

Among the 9449 pathogens isolated, *Escherichia coli* (*n* = 4902, 51.9%) was the most frequently isolated, followed by other *Enterobacteriales* (*n* = 2694, 28.5%), *Enterococcus* spp. (*n* = 995, 10.5%), *Pseudomonas aeruginosa* (*n* = 566, 6%), other Gram-positives (*n* = 231, 2.4%) and other non-fermenting Gram-negatives (*n* = 61, 0.6%). *Proteus mirabilis* (*n* = 988), *Klebsiella pneumoniae* (*n* = 633) and *Klebsiella*
*oxytoca* (*n* = 307) were the most frequent among the other *Enterobacteriales*, and *Enterococcus faecalis* (*n* = 850) was the most frequent among enterococci. A complete list of the isolated pathogens is available in Appendix A. 

### 3.1. Antibiotic Resistant UTI by Pathogen

Data on resistance are expressed as proportions of resistant strains over tested strains. *Escherichia coli* and other *Enterobacteriales* were the most frequently isolated pathogens and presented similar distributions of antibiotic resistance. Among *Escherichia coli*, ampicillin resistance was the most frequent (58.8%), followed by SXT (31%) and AMC (29.9%). As for other *Enterobacteriales*, AMC resistance was 38.3% and SXT was 22.6%. Overall ampicillin resistance was 78.2%, falling to 44.9% among 931 tested *Proteus mirabilis* and reaching nearly 100% among the remaining other *Enterobacteriales*. For *Pseudomonas aeruginosa*, resistance to piperacillin-tazobactam was 8.3%; to ceftazidime, 7.3%; and to ciprofloxacin, 4.3%. Overall ampicillin resistance was 8.3% for the *Enterococcus* spp. group, 76.5% among 98 *Enterococcus faecium* and 0.2% among 832 *Enterococcus faecalis*. Data for specific pathogens and antibiotic resistance are shown in Figure 2 and detailed in Appendix A. 

*Escherichia coli* and other *Enterobacteriales* altogether accounted for 80.4% of all isolates (see Figure 1; 7596 strains in 5190 patients; characteristics are reported in Appendix A), so further analyses were then mainly focused on these groups. AMC and CIF resistance was 32.8% and 13.7%, respectively, and remained quite stable among the different age groups. On the contrary, resistance to CIP and SXT increased with age (Table 1 and Appendix A).

In Table 2, risk factors for isolation of antibiotic resistant *Enterobacteriales* are reported. In general, considering the antibiotics with oral formulations (AMC, CFI, CIP and SXT), similar results were observed for AMC and CFI and for CIP and SXT. Age significantly influenced the risk of resistance to all antibiotics, especially for CIP and SXT, which showed increased risks of resistance with age, whereas for the other drugs, increasing age was a “protective” factor, since compared to children aged <6 months, the risk decreased.

As for the risk of resistance according to the department of admission, the risk increased for all antibiotics and for all departments compared to the Emergency Department. Additionally, the previous number of episodes was associated with an increased risk of resistance for all antibiotics for subjects with at least one previous event compared to patients without a history of UTI.

### 3.2. AMC/CFI and CIP/SXT Resistance Prediction Models

Since among *Enterobacteriales*, the resistance ratios were very similar for AMC and CFI (2466/7351, 33.5%) and for CIP and STX (2304/7577, 30.4%), a predictive model was implemented to evaluate the probability of isolating a strain resistant to these two groups of drugs at the time of urine sampling. Seventy percent (*n* = 5147 for AMC/CFI and *n* = 5305 for CIP/SXT) of the total number of samples was assigned to the training group and 30% (*n* = 2204 for AMC/CFI and *n* = 2272 for CIP/SXT) to the validation group. The results of the final models on the training data set are shown in Table 3 with the weight/point assigned to each one for the score construction. None of the interactions between the variables of the final models were statistically significant. 

The overall predictive model performance was moderate. The Bier scores were 0.216 and 0.199 for AMC/CFI and CIP/SXT, respectively. The predictive accuracy of models was weak for the discrimination performance (the ability of the model to distinguish between events and non-events): For AMC/CFI, the model AUC-ROC was 61.8%, 95% CI (60.1–63.4) in the training cohort; 59.7%, 95% CI (57.2–62.2) in the validation cohort; and 60.9%, 95% CI (59.2–62.1) in the 10-fold cross-validation. For CIP/SXT, AUC-ROC was 64%, 95% CI (62.4–65.6) in the training cohort; 63.3%, 95% CI (60.8–65.8) in the validation cohort; and 63.5%, 95% CI (61.1–64.0) in the 10-fold cross-validation (Appendix A). However, the models achieved very good calibration (how close the predicted probabilities are to the actual rate of events/“real” probability), as shown in Figure 3 by comparing predicted and observed probabilities in the calibration plots. In particular, the observed prevalence of resistance to AMC/CFI in the training cohort was 33.5% (1727/5147), and the average estimated risk given by model was 0.334 (i.e., 33.4%); for CIP/SXT the observed prevalence was 30.4% (1613/5305) and the average estimated risk was 0.304 (i.e., 30.4%).

Table 4 shows the simple risk scoring system and how individual total scores relate to the specific risk of infection due to resistant *Enterobacteriales*. Considering AMC/CFI, those patients (male aged <6 months admitted in Neonatal or Pediatric ICU and with a history of previous episodes) with total scores of 22 had an estimated risk of over 60%. For CIP/SXT, patients with the maximum total score of 27 (patients aged >7 years, in Hematology/Oncology and with a history of previous episodes) had an estimated risk of over 70%.

## 4. Discussion

In the present, retrospective, single center study, the proportions of antibiotic resistance among more than 9000 pathogens isolated from urine sample in children were analyzed. Having a large number of cases, we analyzed not only the proportions of isolated pathogens and their patterns of resistance to selected antibiotics; factors associated with a higher probability of antibiotic resistance among *Enterobacteriales* were identified, especially resistance to antibiotics with oral formulations generally administered for urinary tract infections (AMC, CIF, CIP, SXT). High rates of antibiotic resistances for orally administrated drugs that are commonly prescribed in pediatrics were documented. Overall, *Enterobacteriales* represented about 80% of isolated strains, and resistance to AMC was the highest (near 33%), followed by SXT (28%); the lowest was for CIP, which was detected in just under 10% of isolated pathogens. These results are in line with other recent data from other Italian regions. In a pediatric cohort, under the age of 18 years, in Emilia-Romagna [26], *Enterobacteriales* were the most frequently isolated pathogens (82%), and overall, AMC resistance was 34%. Additionally, in a study of children aged 0 to 6 years in Tuscany [33], *Enterobacteriales* represented the majority of isolates (71%), and isolated uropathogens showed high resistance rates to AMC (26%), particularly in children under one year of age (28%). In this study, a worrisome proportion of resistance in children aged <6 months was observed, especially resistance to AMC (35%). This may be associated with peripartum exposure to maternal antibiotics, which increases the risk of resistant rods in the newborn. Furthermore, the increased resistance rates in children under one year of age could be related to the acquisition of nosocomial microorganisms at the time of birth [33]. In our study, age ≤ 6 months represented a significant risk factor for infection due to strains resistant to AMC/CIF, but not for CIP/SXT, which showed an increasing risk of resistance with increasing age at episode. This observation is particularly worrisome, since this patient group (children ≤ 6 months of age) presented with first UTI episodes due to bacterial strains already resistant to AMC/CIF in a not negligible proportion. These children are frequently affected with urinary tract abnormalities and have a high risk of recurrency for UTIs, and above all vesico-ureteral reflux, and could go through repeated antibiotic courses (prophylaxis and/or therapy), which are a well-known risk factor for antibiotic resistance [11,16]. Our results provide warnings on the feasibility of applying the most recent national recommendations for treatment of the first episode of urinary tract infections in pediatrics, which indicate AMC/CIF as the preferred first-line oral therapy [23]. The choice of use of AMC as an empirical first-line treatment should be carefully evaluated, above all in children under one year of age, and limited to patients in good clinical condition [33]. 

A further important observation is that in any episode observed outside the Emergency Department, in particular, in Hematology/Oncology, Nephrology or Neonatal ICU, there was a significant risk of the presence of a strain resistant to oral antibiotics. This is partially in contrast with our previous study [19], in which, among 4596 positive urine cultures from 2007 to 2014, *E. coli* represented the majority of isolates (3006, 65.4%) and AMC resistance was 30%. In addition, an increased risk of AMC resistance was significantly associated with hospitalization in Nephrology or other wards managing urinary tract malformation with respect to wards not dealing with this malformation. Instead, a more recent work reported high resistance in Hematology and Oncology departments [34].

Finally, our data confirmed an increased risk of isolation of a resistant strains in case patients who had experienced more than one episode of UTI. Considering repeated same-patient cultures, a personal component of memory-like correlations of resistance can be assumed. These correlations can represent recurrent infections with the same strain or correlations with other patient-specific factors that further contribute to the predictability of resistance [19,21]. 

As a secondary aim, an initial scoring system was proposed, mainly to guide the clinical decision-making process for antibiotic treatment and not to distinguish between events and non-events. Since antibiotic resistances in pathogens causing UTIs are emerging, research describing risk factors associated with resistances and developing prediction models is of great interest and will help to better target initial antibiotic treatment. The developed models presented a moderate ability (AUC 60%) to discriminate on the basis of the probabilities of those who had an infection with resistant *Enterobacteriales* from those who did not, and therefore, be able to find the best cut off. However, a well discriminating model may be useless if the decision threshold for clinical decisions is outside the range of predictions provided by the model. Even a poorly discriminating model may be clinically useful if the clinical decision is close to a “toss up.” This implies that the threshold is right in the middle of the distribution of predicted risks, but also that evaluation of calibration is important if model predictions are used to inform patients or physicians to make decisions [29]. Calibration is especially important when the aim is to support decision-making, even when discrimination is moderate [35]. The need for a further evaluations with new data, possibly collected prospectively, is the major limitation of this study, together with its retrospective method that caused a limited number of potential risk factors being included in the statistical model. The data available for this study were extracted from a microbiology laboratory database, so data on patients’ medical history, e.g., antibiotic administration for prophylaxis or treatment of previous UTIs episodes, were not available. However, the department of admission, where the UTIs were diagnosed, played an important role in this analysis, and this factor could be considered as a “proxy variable” for the patient’s clinical condition. This study proposed a weighted scoring model based on simple information available at the time of hospital admission. The easy scoring system also clearly showed that a non-negligible (about 20% and more) risk of resistance to AMC/CIF was present also in patients with no risk factors (i.e., negative or zero scores). This is particularly worrisome if we consider that bacteria causing urinary tract infections are often carried asymptomatically in the human body and are therefore frequently exposed to antibiotics, including those taken to treat other infections, and that AMC is among the most frequently prescribed antibiotics in Italy in pediatrics, and its use can drive also resistance to other antibiotics [36,37,38]. This risk was lower for SXT/CIP, but at least for CIP, use should be restricted not only because of its potential risk of toxicity in pediatrics, but also since it could represent the only oral treatment for *Pseudomonas*
*aeruginosa* or other Gram-negatives resistant to other antibiotics [39,40,41]. 

## 5. Conclusions

Our study provided insight into the predictors associated with a higher probability of antibiotic resistance among *Enterobacteriales*, especially to antibiotics with oral formulations generally administered in urinary tract infections. The developed scoring system could guide the clinical decision-making process for antibiotic treatment, based on four variables that are easy to define in clinical practice at the time of hospital admission. Proper use of this tool could minimize the time required to manage UTI and could reduce workloads and costs. Further validation works on this novel predictor should be perused to guide appropriate antimicrobial therapy and to improve the prognosis for these infections.

## Figures and Tables

**Figure 1 antibiotics-11-00720-f001:**
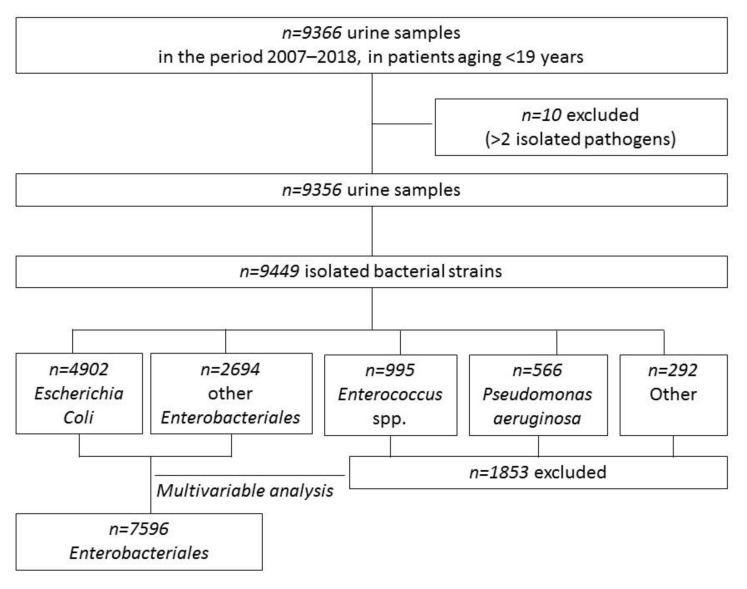
Flow chart of the study of urine samples and isolated pathogens.

**Figure 2 antibiotics-11-00720-f002:**
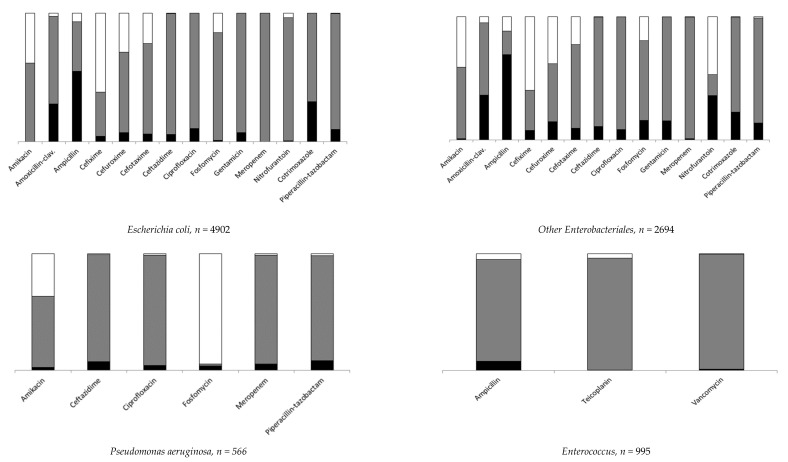
Distribution of antibiotic resistant strains (details in Appendix A). A black box represents the percentage of resistant, a gray box the percentage of susceptible and a white box the percentage of not tested.

**Figure 3 antibiotics-11-00720-f003:**
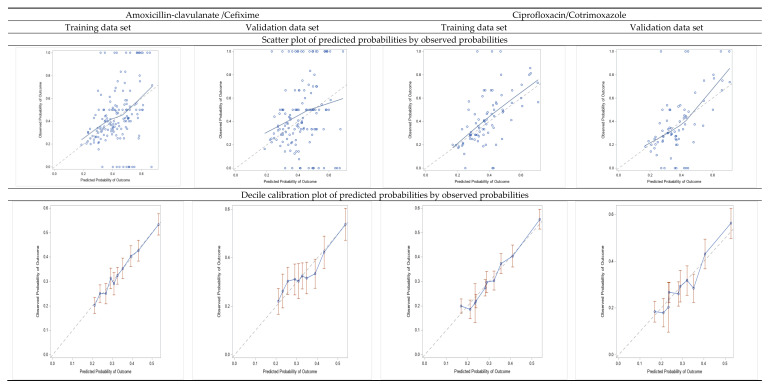
Calibration plots: scatter plots of predicted probabilities by observed probabilities of resistance, and decile calibration plots for the training and validation data sets.

**Table 1 antibiotics-11-00720-t001:** Distribution antibiotic resistance among 4902 *Escherichia coli* and 2694 other *Enterobacteriales* by age at sampling.

	Age at Sampling														
	0–6 Months			7 Months–2 Years			3–7 Years			>7 Years			Total		
Antibiotics	Resistant	Susceptible	Not Tested	Resistant	Susceptible	Not Tested	Resistant	Susceptible	Not Tested	Resistant	Susceptible	Not Tested	Resistant	Susceptible	Not Tested
Amoxicillin-clavulanate	661 (34.8)	1239	45	673 (31.1)	1493	89	483 (33.6)	955	60	592 (32.3)	1242	64	2409 (32.8)	4929	258
Ciprofloxacin	94 (4.9)	1842	9	190 (8.5)	2056	9	177 (11.8)	1317	4	264 (13.9)	1630	4	725 (9.6)	6845	26
Cotrimoxazole	352 (18.2)	1585	8	595 (26.5)	1653	7	488 (32.6)	1007	3	688 (36.3)	1206	4	2123 (28.0)	5451	22
Cefixime	118 (15.7)	635	1192	117 (12.9)	793	1345	70 (11.5)	538	890	102 (14.4)	604	1192	407 (13.7)	2570	4619
Ceftazidime	159 (8.2)	1780	6	172 (7.7)	2072	11	100 (6.7)	1393	5	137 (7.2)	1758	3	568 (7.5)	7003	25
Piperacillin-tazobactam	244 (12.6)	1686	15	247 (11.0)	1992	16	148 (9.9)	1345	5	202 (10.7)	1685	11	841 (11.1)	6708	47

R = resistant; S = susceptible; N = not tested, numbers in parenthesis are proportions (resistant/tested).

**Table 2 antibiotics-11-00720-t002:** Multivariable logistic regression models for resistance to different antibiotics among 4902 *Escherichia coli* and 2694 other *Enterobacteriales*.

	Odds Ratio (95%CI) ^					
Factors	Amoxicillin-Clavulanate *n* = 7338 * R = 32.8%	Cefixime, *n* = 2977 * R = 13.7%	Ciprofloxacin, *n* = 7570 * R = 9.6%	Cotrimoxazole, *n* = 7574 * R = 28.0%	Ceftazidime, *n* = 7571 * R = 7.5%	Piperacillin-Tazobactam, *n* = 7549 * R = 11.1%
Sex, *p*-value	<0.001	0.302	0.988	0.341	0.318	<0.001
Male vs. female	1.4 (1.2–1.6)	1.1 (0.9–1.4)	1.0 (0.8–1.3)	0.9 (0.8–1.1)	1.1 (0.9–1.4)	1.4 (1.2–1.7)
Age at sampling, *p*-value	0.002	<0.001	<0.001	<0.001	<0.001	0.0001
7 months–2 years vs. ≤6 months	0.9 (0.8–1.1)	0.6 (0.5–1.0)	1.8 (1.3–2.4)	1.5 (1.3–1.8)	0.9 (0.7–1.2)	0.9 (0.7–1.1)
3–7 years vs. ≤6 months	0.9 (0.8–1.1)	0.5 (0.4–0.8)	2.2 (1.5–3.1)	1.8 (1.5–2.2)	0.6 (0.4–0.9)	0.7 (0.5–0.9)
>7 years vs. ≤6 months	0.7 (0.6–0.9)	0.5 (0.3–0.8)	2.0 (1.3–2.8)	1.9 (1.5–2.3)	0.5 (0.3–0.7)	0.6 (0.5–0.8)
Department of admission, *p*-value	<0.001	<0.001	<0.001	<0.001	<0.001	<0.001
Nephrology vs. Emergency	2.5 (2.0–3.1)	2.4 (1.7–3.6)	2.2 (1.6–3.3)	1.8 (1.5–2.3)	4.0 (2.8–5.8)	2.9 (2.2–3.8)
Surgery/Orthopedics vs. Emergency	1.5 (1.2–1.9)	2.3 (1.5–3.6)	1.8 (1.2–2.6)	1.1 (0.9–1.3)	3.2 (2.1–4.7)	2.0 (1.5–2.7)
Infectious Diseases vs. Emergency	1.4 (1.1–1.8)	1.4 (0.8–2.4)	2.1 (1.3–3.1)	1.4 (1.1–1.8)	2.0 (1.2–3.2)	1.3 (0.9–1.9)
Hematology/Oncology vs. Emergency	2.5 (1.8–3.5)	5.0 (2.9–8.5)	3.9 (2.5–6.2)	4.7 (3.2–6.8)	7.4 (4.5–12.2)	3.8 (2.6–5.8)
Neonatal Pediatric ICU vs. Emergency	3.1 (2.3–4.3)	2.9 (1.6–5.5)	2.8 (1.6–4.7)	1.4 (1.0–2.0)	5.0 (3.2–7.9)	3.5 (2.4–5.1)
Others vs. Emergency	1.6 (1.4–1.9)	2.2 (1.5–3.1)	1.7 (1.2–2.3)	1.2 (1.0–1.4)	3.5 (2.6–4.8)	2.1 (1.6–2.7)
Previous number of episodes, *p*-value	<0.001	<0.001	<0.001	<0.001	<0.001	0.0004
1 vs. 0	1.2 (1.0–1.3)	1.3 (1.0–1.7)	1.3 (1.1–1.6)	1.2 (1.1–1.4)	1.5 (1.2–1.9)	1.4 (1.1–1.7)
>1 vs. 0	1.4 (1.2–1.7)	1.8 (1.2–2.5)	2.0 (1.5–2.7)	1.5 (1.3–1.8)	1.6 (1.1–2.1)	1.4 (1.1–1.8)

R = resistant. ^ Robust standard error for correction due to possible multiple events per subject. * Not tested were excluded from analysis.

**Table 3 antibiotics-11-00720-t003:** Multivariable logistic regression models of training data sets. Models and points system to predict risk of specific drug-resistant urinary tract infection due to *Enterobacteriales*.

Factors+A1:E28	Odds Ratio (95%CI) ^1^	β	LRT, *p*-Value	Points
Amoxicillin–clavulanate/Cefixime, *n* = 5147,				
Intercept = −1.1136				
Sex			<0.001	
Male vs. female	1.452 (1.261–1.672)	0.3731		4
Age at sampling			0.002	
7 months–2 years vs. ≤6 months	0.862 (0.726–1.024)	−0.148		−2
3–7 years vs. ≤6 months	0.907 (0.738–1.115)	−0.097		−1
>7 years vs. ≤6 months	0.696 (0.560–0.864)	−0.363		−4
Department of admission			<0.001	
Nephrology vs. Emergency	2.287 (1.812–2.887)	0.827		10
Surgery/Orthopedics vs. Emergency	1.555 (1.220–1.981)	0.441		5
Infectious Diseases vs. Emergency	1.321 (0.981–1.780)	0.279		3
Hematology/Oncology vs. Emergency	2.428 (1.719–3.430)	0.887		11
Neonatal or Pediatric ICU vs. Emergency	3.292 (2.346–4.620)	1.192		14
Others vs. Emergency	1.635 (1.340–1.995)	0.492		6
Previous number of episodes			<0.001	
1 vs. 0	1.316 (1.122–1.544)	0.275		3
>1 vs. 0	1.421 (1.158–1.743)	0.351		4
AUC (95% CI) training, *n* = 5147;	0.618 (0.601–0.634)	−	0.17042	−
Brier Score;	0.214			
Spiegelhater z test, *p*-value;	0.989			
Hosmer and Lemeshow test, *p*-value	0.952			
AUC (95% CI) validation, *n* = 2204;	0.597 (0.572–0.622);	−		−
Brier Score;	0.216;			
Spiegelhater z test, *p*-value;	0.988;			
Hosmer and Lemeshow test, *p*-value	0.469			
Range–point total	–	−	−	−4; 22
Ciprofloxacin/ Cotrimoxazole, *n* = 5305,				
Intercept = −1.5882				
Age at sampling			<0.001	
7 months –2 years vs. ≤6 months	1.505 (1.241–1.824)	0.409		4
3–7 years vs. ≤6 months	1.925 (1.561–2.372)	0.655		7
>7 years vs. ≤6 months	1.953 (1.567–2.433)	0.669		7
Department of admission			<0.001	
Nephrology vs. Emergency	1.900 (1.494–2.416)	0.642		7
Surgery/Orthopedics vs. Emergency	1.154 (0.905–1.470)	0.143		2
Infectious Diseases vs. Emergency	1.448 (1.064–1.970)	0.370		4
Hematology/Oncology vs. Emergency	3.896 (2.585–5.872)	1.360		15
Neonatal or Pediatric ICU vs. Emergency	1.503 (1.022–2.211)	0.408		4
Others vs. Emergency	1.290 (1.050–1.583)	0.254		3
Previous number of episodes			<0.001	
1 vs. 0	1.261 (1.077–1.476)	0.232		3
>1 vs. 0	1.568 (1.304–1.886)	0.450		5
AUC (95% CI) training, *n* = 5305;	0.640 (0.624–0.656)	−	0.6472 ^2^	
Brier score;	0.199			
Spiegelhater z test, *p*-value;	0.949			
Hosmer and Lemeshow test, *p*-value	0.189			
AUC (95% CI) validation, *n* = 2272;	0.633 (0.608–0.658)	−		
Brier Score;	0.198			
Spiegelhater z test, *p*-value;	0.972			
Hosmer and Lemeshow test, *p*-value	0.637			
Range–point total	–	−	−	0; 27

The reference factor has a score of zero; LRT = Likelihood-ratio test; AUC = area under the receiver operating characteristic curve; ^1^ robust standard error for correction due to multiple event per subject; ^2^
*p*-value of Chi-square test to compare the areas under two independent ROC curves (training and validation).

**Table 4 antibiotics-11-00720-t004:** Points system and risk estimates of isolating specific drug-resistant *Enterobacteriales* based on multivariable models reported in Table 3.

Point Total	Amoxicillin-Clavulanate/Cefixime %Estimate of Risk	Ciprofloxacin/Cotrimoxazole %Estimate of Risk
−4	19.0	-
−3	20.4	-
−2	21.8	-
−1	23.2	-
0	24.7	17.0
1	26.3	18.3
2	27.9	19.7
3	29.7	21.2
4	31.4	22.7
5	33.2	24.3
6	35.1	26.1
7	37.0	27.9
8	39.0	29.7
9	41.0	31.6
10	43.0	33.6
11	45.1	35.7
12	47.2	37.8
13	49.2	40.0
14	51.3	42.2
15	53.4	44.4
16	55.5	46.7
17	57.5	48.9
18	59.5	51.2
19	61.5	53.5
20	63.5	55.7
21	65.4	58.0
22	67.3	60.2
23	-	62.3
24	-	64.4
25	-	66.5
26	-	68.5
27	-	70.4

## Data Availability

Data are contained within the article or Appendix A. They are available on request from the corresponding author.

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
