# Peer review of "Epidemiology of Antibiotic Resistant Pathogens in Pediatric Urinary Tract Infections as a Tool to Develop a Prediction Model for Early Detection of Drug-Specific Resistance"

_antibiotics, 2022, doi:10.3390/antibiotics11060720_

Round 1

Reviewer 1 Report

Congratulations to the authors for this interesting work, with an impressive amount of statistical data and a very large number of tested strains.

In this manuscript the authors conducted a retrospective analysis on the susceptibility of more than 9400 UTI pathogens isolated  in children during 2007-2018 and identified the risk factors of resistance to different classes of antibiotics.

Moreover, due to the due to the need to administer appropriate antibiotherapy in prophylaxis or therapy, they developed a prediction model that can be helpful in avoiding inefficient antibiotherapy or toxicity (when using broad spectrum antibiotics) 

The authors compared very briefly their recent data with a previous work, without mentioning the period and other relevant features.

It may be helpful to compare the pattern of resistance of UTI pathogens with the data from other regions (at least in Italy)  to assess whether the prediction score can be extended to other areas.

Author Response

Response to Reviewer 1 Comments - manuscript No.: 1713513

Point 1: … The authors compared very briefly their recent data with a previous work, without mentioning the period and other relevant features.

Response 1: We thank the Reviewer for the positive feedback and we agree that relevant features of the our previous work should be summarized.  A sentence has been added in the discussion (line 46-51 in discussion in the track changes file)

Point 2: It may be helpful to compare the pattern of resistance of UTI pathogens with the data from other regions (at least in Italy)  to assess whether the prediction score can be extended to other areas.

Response 2: We thank the Reviewer for this important suggestion which will be developed in a next prospective study to validate the prediction score. However, we agree that it is helpful to compare the pattern of resistance of UTI pathogens with the data from other regions in Italy. A sentence has been added in the discussion (line 18-23 in discussion in the track changes file).

Reviewer 2 Report

The current work titled 'Epidemiology of antibiotic-resistant pathogens in pediatric urinary tract infections as a tool to develop a prediction model for early detection of drug-specific resistance' proposed a prediction model to guide the clinical decision-making process for antibiotic treatment. While the topic is very interesting and significant, the outcomes of the prediction model authors used, may not be sufficient to be published in a good journal like Antibiotics. As the authors mentioned, the predictive accuracy of models was weak for the discrimination performance (the ability of the model to distinguish between events and non-events.

Author Response

Response to Reviewer 2 Comments - manuscript No.: 1713513

Point 1: …While the topic is very interesting and significant, the outcomes of the prediction model authors used, may not be sufficient to be published in a good journal like Antibiotics. As the authors mentioned, the predictive accuracy of models was weak for the discrimination performance (the ability of the model to distinguish between events and non-events

Response 1: We thank the Reviewer for the comment, we agree to clarify  this topic. In this study, the aim of the prediction model was focused to give a possible useful reference to guide clinical practice in treatment decision making and not to distinguish between event and non-event. The very good calibration, that model achieved, supported this aim (Van Calster B, et al. doi: 10.1186/s12916-019-1466-7, Steyerberg EW, et al. doi: 10.1097/EDE.0b013e3181c30fb2). Lines 61-63 and 69-70 in discussion in the track changes file have been changed.

Reviewer 3 Report

The manuscript by Francesca Bagnasco et al. describes the development and validation of a predictive model for the early detection of antibiotic resistance.

This manuscript deserves many revisions before it is eventually considered for publication.

First, because of the critical impact of microbiology, it is important to consider a medical microbiologist in the list of authors.

Overall, prefer passive voice; numbers less than twelve should be capitalized; italicize "i.e."/"vs."/"e.g."

Methods: 

Urine contaminations could not be considered positive urocultures because they only described inappropriate decontamination. (line 75)

How was the number of patients to be included in each cohort determined? Because some of the results might suggest a lack of power, it is crucial to answer this question.

Why did the authors consider a threshold of 100,000 CFU/ml? Thresholds depend on the bacteria and gender of the patients, this needs to be refined.

Blood agar/MacConkey needs to be described appropriately (e.g., manufacturer contact information).

Why did the authors not consider tazocillin and Ceftriaxon for Enterobacterales? 

How did the authors consider the risk of multiple tests they took?

There is a lot of data and it is not limited to administrative data, so the authors should at least inform the patients included.

Results: 

A flow chart for their sample selection would be helpful to highlight the inclusion process.

The description of ampicillin resistance in Enterobacteriaceae is biased by the representation of Enterobacteriaceae in group 2 or 3 for example. This needs to be changed.

Were the enterococci E. faecium or E. faecalis? The implications are different for ampicillin resistance.

More than just a description (lines 180 - 187), the authors should consider giving an explanation for these observations.

Table 3: The authors need to further discuss the impact of admission service, as some differences are obvious (ED vs surgery) but others are not (nephrology vs ED). Also, comparison between patients with 1 and patients with >1 isolated pathogen (>1=2, right?) would be interesting.

Round 2

Reviewer 3 Report

The manuscript has been revised according to my previsous comments.